# Reproducibility study:
# Generative causal explanations of black-box classifiers

## Summary

**Scope of reproducibility**

In the paper by O'Shaughnessy et al. [1], the authors claim to have created a conceptual framework that is able to explain black-box classifiers using latent variables and an appropriate generative model. They also claim that this framework works for *any* classifier and that the latent variables are disentangled (the latent variable describe independent features).

**Methodology**

In this report, we re-implement their code and aim to generate results similar to theirs. Additionally, we extent their research by testing the framework on new datasets. These datasets are the complete MNIST set [2] (in constrast to just the digits 3 and 8) and the CIFAR-10 dataset [3]. Finally, we purposefully used an poorly performing model to discover if the framework is able to find out why the model struggles.

**Results**

It was possible to reproduce the results presented in the paper. Furthermore, the framework was able to highlight the cause of the poor performance of the purposefully poorly performing model. When applying their framework to a more complex dataset, the framework proved to be effective.

**What was easy?**

Although the original code did not work immediately once the code was fixed it was easy to get similar results.

**What was difficult?**

The available code of the original authors did not work completely. There where some issues with broken references and lengthy code which could be hard to understand therefore reusing parts and rewriting others was more time efficient than debugging the available code. Furthermore understanding algorithm 1 in their paper was difficult as many details were left out, such as how to choose an appropriate step size.

**Communication with original authors**

We have not communicated with the original authors.

# 1 Introduction

The ongoing research in the field of machine learning allows the creation of better and more innovative deep learning architectures. However, these advances can lead to increased model complexity, making it harder to explain the behavior of a classifiers.

Gaining insight into causal processes in deep learning models, which often tend to be applied as black-box solutions, will not only increase usefulness in industry applications where transparency is valued or legally required, but also potentially allow machine learning researchers and engineers to better visualize, understand and further develop more robust models.

Some methods that create post-hoc explanations have been constructed, using decision trees [4] or using saliency maps [5]. In contrast to these methods that aim to find explanations using the input values, O'Shaughnessy et al. [1] propose a framework that can explain a models behaviour using a low number of latent variables that represent the data. These latent variables describe features of the input data that can be used to explain the classifiers decision. In this report, we aim to reproduce the results that O'Shaughnessy et al. found and analyse their claims and conclusions. Additionally, we extent their research by evaluating their framework on new models and datasets.

The proposed framework uses a generative network to recreate the dataset using a set of latent variables. As these latent variables describe the data, changing certain variables, the causal factors $\alpha \in \mathbb{R}^K$ , should change the decision of the classifier. Changing the other variables, the non-causal factors $\beta \in \mathbb{R}^L$ that are needed to accurately describe the dataset (e.g. thickness of MNIST digits), should not influence the output of the classifier. The loss function of the generator consists of a term that maximises the similarity between the modelled distribution and the data distribution, as well as a term that maximizes the influence of $\alpha$ on the classifier output. The relative importance of the similarity term can be adjusted with a value $\lambda$.

The main claim of the paper is that the proposed framework is able to explain the decisions of the classifier by observing how the prediction changes based on adjustments of the causal factors $\alpha$. Additionally, they claim that this framework works on *all* classifiers.

As the original authors propose a *conceptual* framework, recreating this framework with a different implementation should be possible. Therefore, most of the code base necessary was recreated. Even though the original code is available, re-implementing proved more time efficient than debugging and understanding the original code. With the re-implemented code, we aim to reproduce the results the original authors got using the framework on the MNIST dataset, using the digits 3 and 8, (which can be seen in figure 1) and the results of the framework on the Fashion MNIST [6] with the classes tops, dresses and coats.

Additionally, we extend the research by examining the framework's ability to find causal relationships in a poorly performing classifier model. If proven effective, the framework could be used to, for example, analyze overfit models to find the features that the classifier overfit on, therefore finding and displaying biases in the model, or the dataset itself.

Finally, we test the framework on more complex datasets than those used in the original paper. These datasets are the complete MNIST dataset [2] (as opposed to just using the digits 3 and 8), and the CIFAR-10 dataset [3]. As the authors claim that their framework works on all models, the framework should still work on models designed to classify more complex datasets. The re-implemented code is available on our GitHub repository[1].

# 2 Method

This section presents all experiments that are conducted to reproduce and extent the research of O'Shaughnessy et al. The hardware used to do these experiments, ranges from notebooks to a computer cluster. The only factor affected by these differences in hardware should be the time needed to train and test the models.

## 2.1 Re-implementing the code

To run the experiments, most of the code base necessary was recreated. The author's code base was updated after we've finished most of the re-implementation so for the purposes of this report we consider the old version (as of 05/01/2021) as our reference point. To train the classifiers, a simple loop was written using PyTorch that continuously uses stochastic gradient descent to optimize the model. The loop is written in such a way that it is model- and dataset-agnostic, allowing us to train all classifiers needed for all experiments that will be conducted for this study. As the method for training the generative model is more complex, PyTorch Lightning was used for the training and debugging process, as well as the

---

[1]https://github.com/FrisoVerweij/FACT

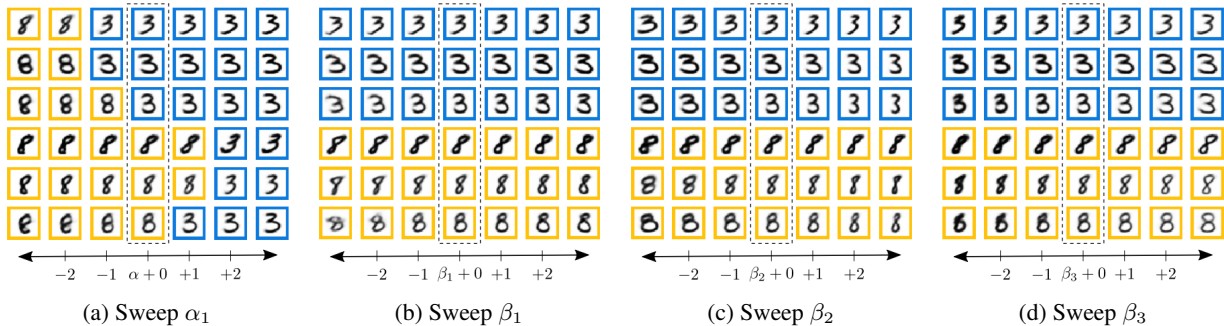

|     |     |     |     |
|-----|-----|-----|-----|
| (a) Sweep $\alpha_1$ | (b) Sweep $\beta_1$ | (c) Sweep $\beta_2$ | (d) Sweep $\beta_3$ |

Figure 1: Figure 3 from the original paper [1]. Each row represents a sample. The y-axis represents change in the given latent variable while keeping all others fixed. We can observe that changing $\alpha$ (a) results in a different output of the classifier, which is denoted by the color of the border. $\alpha$, therefore, is a causal factor that explains the output of the classifier. The opposite is visible in (b), (c), (d), where changing the non-causal factors $\beta$ do not result in different output of the classifier.

Adam optimizer [7]. There was no need to re-implement the calculation process of the causal term used to optimize the generative model, as the framework uses a specific causal influence metric.

In the paper, the authors also present Algorithm 1 that is used to find optimal values for $K, L$ and $\lambda$. Although the algorithm was re-implemented, using it proved too time consuming as it requires a large number of re-runs of the training procedure. If the problem is more complex, both the training time and the number of possible combinations of hyperparameters increase. Therefore, the hyperparameters from the original paper were used or slightly adjusted in case the experiment involved a different problem.

Finally, all code that is used to prepare and load the datasets has been re-implemented.

## 2.2 MNIST with digits 3 and 8

To create figure 1, the authors used a convolutional neural network (CNN) to classify the digits 3 and 8 of the MNIST dataset. The chosen generative model was a convolutional variational autoencoder (CVAE). Their latent distribution was designed to have a single $\alpha$-value ($K = 1$) and 7 $\beta$-values ($L = 7$). For $\lambda$, a value of 0.05 was chosen. As we can see in figure 1, the framework succeeds. Changing $\alpha$ changes the output of the classifier, while changing $\beta$ does not.

To reproduce this figure, the architecture of both models were copied. All hyperparameters were the same as those described in the original paper, apart from the number of training epochs for the CVAE. Instead of training for 43 epochs, the model trained for 30 epochs. Instead of downloading the dataset directly, the library torchvision.datasets was used, allowing us to download the MNIST dataset in a format suited for the trainloop using PyTorch. However, these datasets are not exactly the same, as the original authors used an MNIST dataset where the digits were black on a white background, while the MNIST dataset used in this study uses white digits in a black background. This change, however, should not influence the framework, as the information contained in the images is still the same. Following the training process, a number of latent samples are generated, whose values are adjusted to see the response of the classifier.

## 2.3 Complete MNIST dataset

A logical next step after reproducing figure 1 using just the digits 3 and 8, would be to do the previous experiment using the entire MNIST dataset. The model architectures for classifying and generating the data remained as before, as both the CNN and CVAE architecture are sufficiently complex to classify and generate this relatively simple dataset. As this dataset now has 10 classes, $K$ was increased to 3, allowing the framework to model more causal factors. $L$ remained the same as before, making the number latent distribution consist out of 10 variables. $\lambda$ was increased to 0.1, which makes the generative model focus more on modelling the now more complex data distribution. All other hyperparameters also remained as before.

## 2.4 Fashion MNIST with tops, dresses and coats

The second result we aim to reproduce is the result the original authors got after training the framework on a subset of the Fashion MNIST dataset, a part of which is shown in figure 2. The full figure is available in appendix B.3. This

subsection includes images with the classes *t-shirt/top, dress* and *coat*. The classifier and generative model architecture is the same as those used by the authors, which is also the same architecture used for the previous experiments, regarding the MNIST dataset.

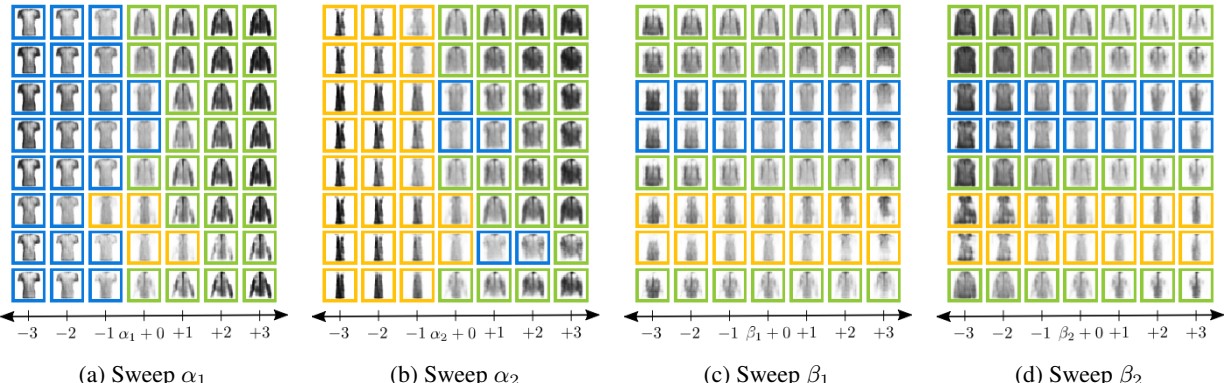

| (a) Sweep $\alpha_1$ | (b) Sweep $\alpha_2$ | (c) Sweep $\beta_1$ | (d) Sweep $\beta_2$ |

Figure 2: A subsection of figure 17 from the original paper [1]. This figure shows how the generated samples result in different images when altering a single latent value. In this figure it can be seen that changing the $\alpha$-values results in a different classification by the classifier, while changing the $\beta$-values does not.

## 2.5 Poor model analysis

Because the framework can be used to explain why a classifier made a certain decision, it should be able to explain and visualize why a poor model yields a low performance in terms of test accuracy. To examine this, we purposefully created a poorly performing classifier that only uses the average pixel intensity values of an image. Again, for this experiment, the MNIST dataset was used, where we only considered digits 3 and 8. The classifier has a simple linear layer that uses the average intensity of the image as input and transforms it into two output values, one for each possible class. After applying softmax, the probability for each class is generated. For the generative model we used the same CVAE architecture as for the other MNIST models.

Because the model uses just a single attribute as input, a single $\alpha$-value would be enough to make the causal effect visible. To model the data distribution, 7 $\beta$-values are used. To make sure the cause of the poorly performing classifier is highlighted, the relative importance of the causal term is increased by decreasing $\lambda$ to 0.001. After generating an image that showcases the effect of changing the $\alpha$-value on randomly generated samples, we would expect the samples changing from very dark images to very bright images (or vice versa). All unnamed hyperparameters remain as before.

## 2.6 The framework on the CIFAR-10 dataset

The MNIST and Fashion MNIST datasets are fairly non-complex datasets, which are suited for a proof of concept. In practice, however, more complex data is often used and although something seems to work fine on less complex data, increasing complexity might lead to unexpected results. For this reason we decided to test the authors framework and loss function on the CIFAR-10 dataset [3] which consists of 32 x 32 colored images. The dataset could also be downloaded from torchvision.datasets which made implementing it with the framework we already had a lot easier.

For the generative model on the CIFAR-10 dataset, a more complex model was needed than the one used on the MNIST and Fashion MNIST datasets. After testing and altering several models [8; 9; 10] we designed our own architecture inspired by these sources. This architecture is specified in Appendix A.1.

For the classifier on just two classes, using the same classifier as for MNIST was sufficient. However, on multiple classes the classifier did not reach the desired accuracy, which is why we used a pretrained VVG-11 [11] model. This model is obtained from [12].

We trained our model on 2 classes, as well as all the classes of the CIFAR-10 dataset. For the full dataset, we kept the total number of latent variables at 64 while varying the $\alpha$-value between 1 and 4 with $\lambda = 0.5$. Additionally, we also kept the $\alpha$-value at 1 with different values for $\lambda$, namely $\lambda = 0.5$, $\lambda = 0.275$, $\lambda = 0.05$. The remaining hyperparameters were kept the same as for the MNIST and Fashion MNIST datasets. For the classifier, we quickly trained the model for 3 epochs, starting with the pretrained weights, with a decreased learning rate of 0.001 to make sure that it was performing as desired.

## 3  Results and analysis

### 3.1  MNIST with digits 3 and 8

The results of the attempt to reproduce figure 1 are shown in figure 3. Looking at this figure, we can see that a change in $\alpha_1$ corresponds to a change of the output of the classifier. For each sample in subfigure (a), the classifier changed its decision after the digits changed shape. Looking at subfigures (b), (c) and (d), we see that changing $\beta_1, \beta_2$ or $\beta_3$, does not result in a different output of the classifier. Looking at (b) and (d), we can see that changing $\beta_1$ changes the width of the digit, while changing $\beta_3$ changes the thickness of the digit. Even though the specific latent factors $\beta$ encode for different attributes when compared to the original figure, we consider this a successful reproduction. Additional results, displaying all latent factors, can be found in appendix B.1.

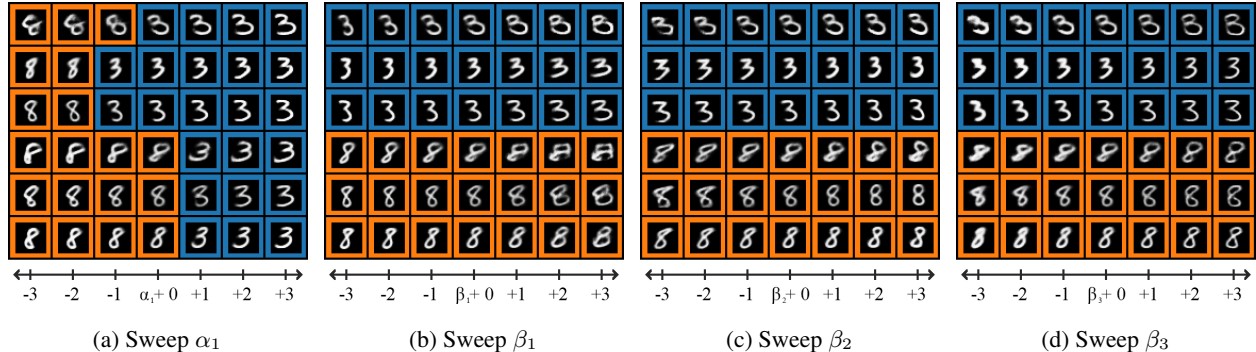

(a) Sweep $\alpha_1$  (b) Sweep $\beta_1$  (c) Sweep $\beta_2$  (d) Sweep $\beta_3$

Figure 3: The reproduced figure 1. In each subfigure, one latent factor is varied, while all others are fixed. Each row represents one sample. The color of the border indicates the decision of the classifier.

### 3.2  Complete MNIST dataset

After training the same CNN and CVAE architecture on the complete MNIST dataset, generating samples and varying the latent variables resulted in figure 4. This figure shows the sweeping process of four of the ten latent variables. Subfigure (a) shows that changing $\alpha_2$ again results in different classifications. This time, as there are multiple classes, we can see that changing $\alpha_2$ can change the prediction of the classifier multiple times. Subfigures (b), (c) and (d) show that changing the $\beta$-values barely affects the classification process. We can see that the digit 3 is consistently classified the same as the 8. However, this is most probably due to a bad sample for the digit 3. Looking at the effect of changing the $\beta$-values on the generation process, we can see a clear change in angle, thickness and width for $\beta_3, \beta_4$ and $\beta_6$, respectively. Therefore, the framework seems to perform well on this more complex dataset. Additional results regarding the complete MNIST dataset can be found in appendix B.2.

### 3.3  Fashion MNIST with tops, dresses and coats

The effect of changing the latent variables are visible in figure 5. Looking at subfigures, we can see that when changing the $\alpha$-values, the classifications change. In contrast, changing the $\beta$-values change some attributes of the shapes, such as width at the bottom of the image, but do not change the decision of the classifier. This is in line with the original authors findings. The full figure is available in appendix B.3.

### 3.4  Poor model analysis

After training the framework, the effect of changing the $\alpha$-value is visible in figure 6. As expected, changing the $\alpha$ changes the average intensity of the image, which is exactly the only input that the classifier uses. We can see that the classifier works as intended, as all bright images are marked with an orange border, while all dark images are marked with a blue border. We can see that latent factor $\beta_1$ and $\beta_2$ seem to encode for digit brightness and/or thickness as well, but this might be because the causal term is designed to maximise the causal influence of the $\alpha$-factors, but not to minimize the $\beta$-factors. The result of changing all latent factors not shown in figure 6 are presented in appendix B.4

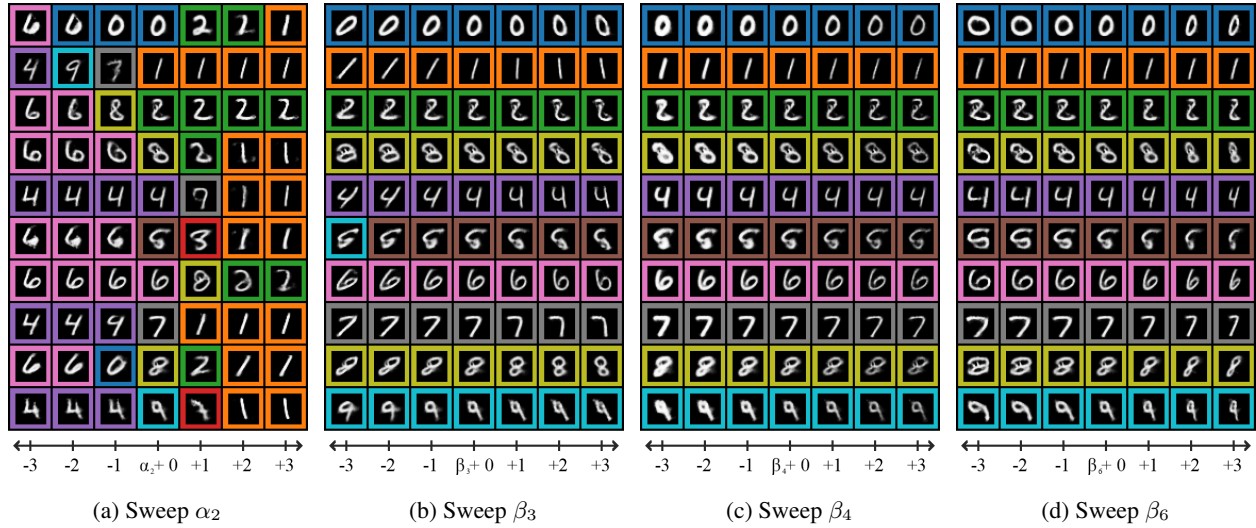

|  |  |  |  |  |  |  |
|---|---|---|---|---|---|---|
| -3 | -2 | -1 | $\alpha_2$+ 0 | +1 | +2 | +3 |
| (a) Sweep $\alpha_2$ | | | (b) Sweep $\beta_3$ | | | (c) Sweep $\beta_4$ | | | (d) Sweep $\beta_6$ |

Figure 4: Each subfigure show the results of changing a single latent factor, while keeping the others fixed, on the complete MNIST dataset.

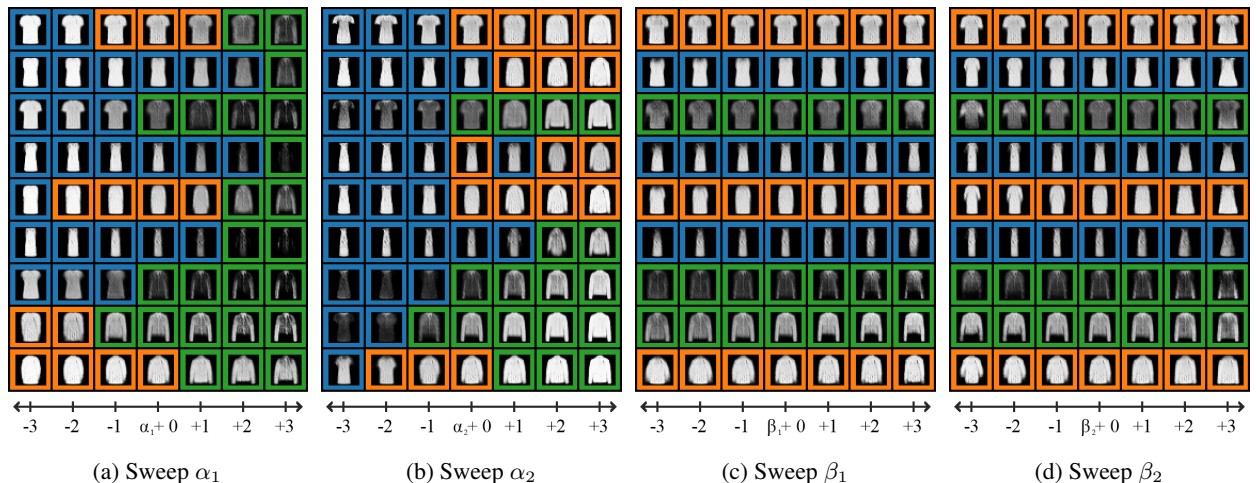

(a) Sweep $\alpha_1$   (b) Sweep $\alpha_2$   (c) Sweep $\beta_1$   (d) Sweep $\beta_2$

Figure 5: The results of the Fashion MNIST experiment.

## 3.5 The framework on the CIFAR-10 dataset

When training on just two classes, the causal term seems so have the desired effect, which can be seen in Figure 13 in appendix B.5. Here, we can can clearly see the output of the classifier change when we vary alpha while the output remains the same when we vary beta. In this example it seems to be the case that the colors red and blue influence the classifier output the most and therefore alpha encodes these colors. One issue is that the generated images are not very detailed which is especially important when testing on more complex generated images.

Even though the CVAE was a lot more complex than before, the images did not reach the visual quality that we desired in order to test the extent of the authors framework. When training on the full CIFAR-10 dataset we saw that the generated images were a lot better. This was the case for training the CVAE without and with the causal term. The increase in quality is likely caused by having more training samples and/or having more diverse images which results in more detailed images for all the classes. Therefore, we decided to abandon the CIFAR-10 dataset with just two classes and focus on the full dataset. The results for the full dataset with one $\alpha$ and $\lambda = 0.5$ are shown in figure 7. Here we can clearly see that the colors red and blue seem to be encoded in the $\alpha$-variable. Besides these colors we can also see a lot of artifacts occuring in the images. This makes sense as generating such an artifact is an effective way of creating causality between $\alpha$ and the classifier. Even when we increase the number of $\alpha$-values the artifacts remain in most of

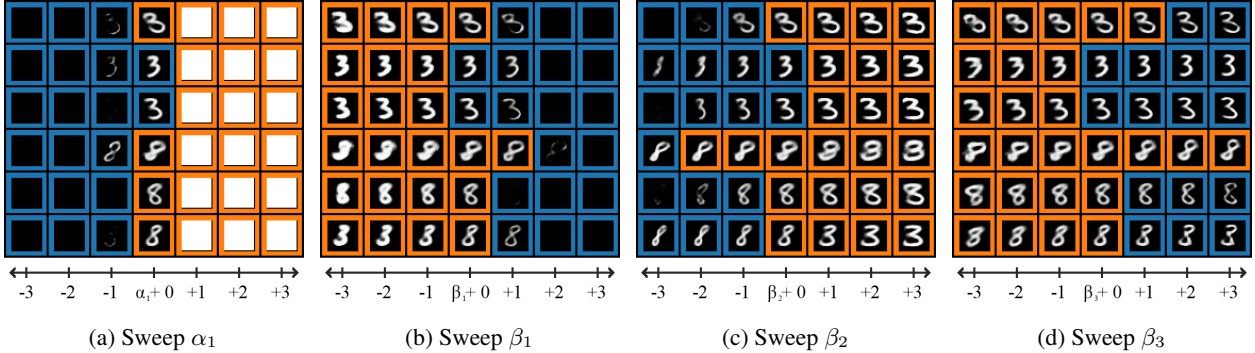

(a) Sweep $\alpha_1$      (b) Sweep $\beta_1$      (c) Sweep $\beta_2$      (d) Sweep $\beta_3$

Figure 6: The results of the poor model analysis. In subfigure (a), $\alpha_1$ encodes for the image brightness. It is important to note though, that $\beta_1$ and $\beta_2$ seem to do the same.

the images and are often visible in the first few epochs. When looking at the loss curves we can also see that the lower the causal loss becomes the more the artifacts become visible even before it converges. Changes $\lambda$ also does not make much of a difference as we found that $\lambda$ mainly controls the rate with what the causal loss drops which is illustrated in Figures 14 and 15 in appendix B.5.

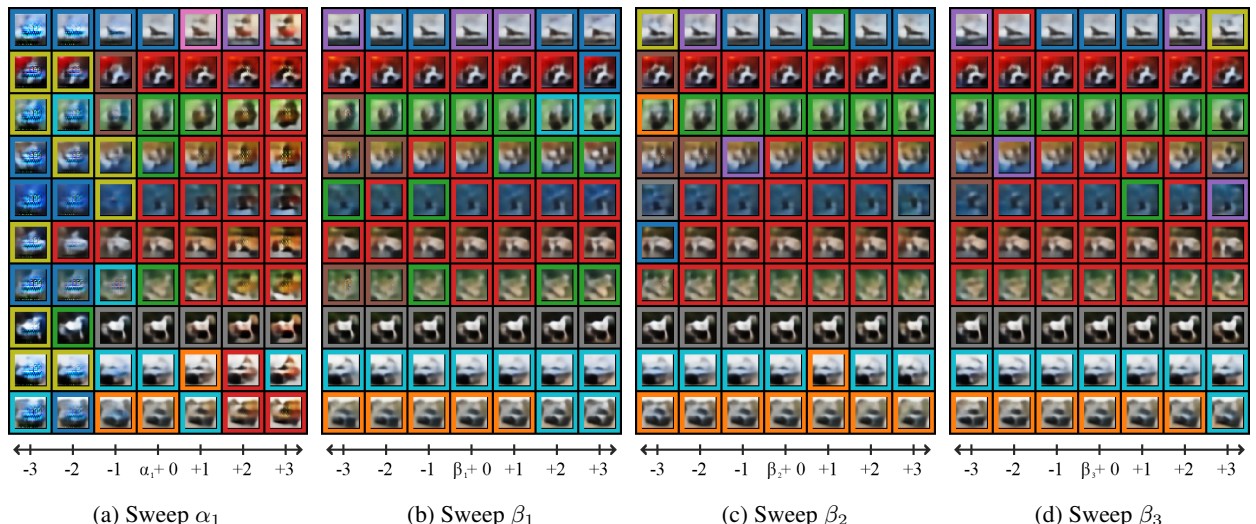

(a) Sweep $\alpha_1$      (b) Sweep $\beta_1$      (c) Sweep $\beta_2$      (d) Sweep $\beta_3$

Figure 7: The results of the CIFAR-10 dataset with one $\alpha$-value and $\lambda = 0.5$.

## 4 Discussion

### 4.1 Overall Reproducibility

After obtaining the results, we can conclude that we were able to reproduce both figure 1 and 2 that were produced by the O'Shaughnessy et al. Additionally, we found that the framework still succeeds when using the complete MNIST dataset. As expected, the framework was able to highlight the problem of the poorly performing model, as the model showed that changing the $\alpha$-value, changed the average pixel intensity of the image. When testing the framework on the CIFAR-10 dataset the results were harder to interpret. It was possible to get a latent space with causal and non-causal dimensions but it was harder to interpret these dimensions.

### 4.2 Issues Encountered

Because we experimented with different datasets and even different models, hyperparameters greatly influence the results, which was an issue we faced, as the algorithm provided by the authors to find the optimal hyperparameters was not entirely clear and is not very feasible for larger models due to limited computational resources.

205 Another issue we faced and discussed was actually quantifying the quality of the encoder-decoder model results.
206 Because there isn't a concrete metric for disentanglement of $\alpha$ and $\beta$, we often relied on our intuition to judge the
207 quality of results and whether the $\beta$ factors really did not overlap with the features that $\alpha$ controlled, and vice versa.
208 This approach is of course prone to personal bias and interpretations of certain patterns that might have arised from a
209 good random result. Nevertheless, experimentation lead us to reproduce the paper's results and even expand them by
210 additional experiments.

211 For the CIFAR-10 dataset a lot more issues arised especially surrounding the explanability of the causal factors as we
212 saw in section 3.5.

### 4.3 Usefulness of Explanations

214 The proposed method does indeed produce visual representations of causal factors that are disentangled from the
215 non-causal factors and is therefore able to provide a explanation and gives insight into the model itself. However, we
216 question the usefulness of the visual explanation of the latent factors $\alpha$ when used on more complex datasets. Looking at
217 Figure 8 and other examples, we can see that there are disentagled causal and non-causal factors and using intuition we
218 could even derive verbal explanations such as 'thickness' or 'sharpness' being non-causal, furthermore we can see that
219 the the transition between a '3' and an '8' is consistent with intuition that the right hand side of an '8' has to be removed
220 to get a '3'. However when we look at $\alpha_2$ from Figure 3.2 we do not know how to interpret that latent dimension
221 because the generated images just seem to change from one digits to the other without a clear human interpretable
222 transition between the digits. Another issue is with the full CIFAR-10 dataset in which artifacts arise in the causal latent
223 dimensions. On the one hand this could be a good explanation, highlighting that the classifier is not as good as we
224 thought because it is heavily influenced by those artifacts, on the other hand this could be an anomaly that arises from
225 the framework itself. More research is needed to find out why these artifacts arise.

226 Overall it seems that for simple models this framework works as one would hope for and generates sensible explanations
227 but it seems not so straightforward to adopt this technique to more complex datasets.

### 4.4 Feasibility of Hyperparameter Optimization

229 Although not the central part of the paper, Algorithm 1 was used to find the optimal hyperparameters of $K$, $L$, and
230 $\lambda$. The issue we see is that even with low granularity of increments, the algorithm still necessitates tens of re-runs of
231 the encoder-decoder training process, which can be unfeasible for big state of the art models working on datasets with
232 hundreds of gigabytes of complex multi-dimensional data. Additionally, it is likely that the encoder-decoder model
233 itself has to be tweaked and possibly enlarged when working on more complex data and classifiers such as with the
234 CIFAR-10 dataset, therefore further multiplying the computational resources needed. It would be useful to find sets of
235 initial hyperparameters or $\alpha$ / $\beta$ ratios for certain types of datasets or models that performs decently well so that the
236 process converges faster.

### 4.5 Causal Influence

238 In the original paper they present the loss function of this method as

$$\underset{g \epsilon G}{\arg\max}\, C(\alpha, Y) + \lambda \cdot D(p(g(\alpha, \beta)), p(X)) \tag{1}$$

239 where $D(p(g(\alpha, \beta)), p(X))$ is a measurement of similarity between the generated image and the sample, and $C(\alpha, Y)$
240 is the metric of causal influence of $\alpha$ on $Y$. In this function, the causal term maximizes the causal influence of $\alpha$
241 while the $\beta$ is only used in the similarity term. However, this means that $\beta$ factors can also have a causal effect if the
242 dimensionality of $\alpha$ is insufficient to cover all the causal features, because it's causal effect is only minimized implicitly.
243 In further research it would be interesting to introduce a term that explicitly minimizes the causal influence of $\beta$ to
244 make sure that only $\alpha$ has causal influence.

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

## Appendix A: Used models and datasets

### A.1: Model architectures

In this section, all model architectures are specified. The classifier architecture used in the experiments using the MNIST or Fashion MNIST is shown in table 1 and the architecture used for the poor model analysis is displayed in table 2.

The architecture of the generative model used in the MNIST and Fashion MNIST datasets are presented in table 3. The architecture of the generative mdel used in the experiment using the CIFAR-10 dataset is shown in table 4.

| Architecture for MNIST and Fashion MNIST |
|:---:|
| Input (28×28) |
| Conv2d (32 channels, 3×3 kernels, stride 1, pad 0) |
| ReLU |
| Conv2d (64 channels, 3×3 kernels, stride 1, pad 0) |
| ReLU |
| MaxPool (2×2 kernel) |
| Dropout (p = 0.5) |
| Linear (128 units) |
| ReLU |
| Dropout (p = 0.5) |
| Linear (M units) |
| Softmax |

Table 1: The classifier architecture used for all experiments using either the MNIST dataset or the Fashion MNIST dataset.

| Architecture for the poor model analysis |
|:---:|
| Input (28×28) |
| average over each pixel value (hard coded) |
| Linear(2) |
| Softmax |

Table 2: The classifier architecture used for the poor model analysis. This model is a mixture of a hard coded model and a neural net

| Architecture of the generative model used in the MNIST and Fashion MNIST experiments | |
|:---:|:---:|
| Encoder architecture | Decoder architecture |
| Input (28×28) | Input (K + L) |
| Conv2 (64 chan., 4×4 kernels, stride 2, pad 1) | Linear (3136 units) |
| ReLU | ReLU |
| Conv2 (64 chan., 4×4 kernels, stride 2, pad 1) | Conv2Transp (64 chan., 4×4 kernels, stride 1, pad 1) |
| ReLU | ReLU |
| Conv2 (64 chan., 4×4 kernels, stride 1, pad 0) | Conv2Transp (64 chan., 4×4 kernels, stride 2, pad 2) |
| ReLU | ReLU |
| Linear (K + L units for both $\mu$ and $\sigma$) | Conv2Transp (1 chan., 4×4 kernel, stride 2, pad 1) |
| | Sigmoid |

Table 3: The architecture of both the encoder and decoder that make up the CVAE used in the MNIST and Fashion MNIST experiments.

| Architecture of the generative model used in the CIFAR-10 experiments | |
|---|---|
| Encoder architecture | Decoder architecture |
| Input 32x32 | Input K+L |
| Conv2 (128 chan., 3×3 kernels, stride 2, pad 1) | Linear(512) |
| ReLU | ReLU |
| Conv2 (128 chan., 3×3 kernels, stride 1, pad 1) | Linear(2*16*256) |
| ReLU | ReLU |
| Conv2 (256 chan., 4×4 kernels, stride 2, pad 1) | Conv2Transp(256, 3x3 kernels, stride 2, padding 1, output_padding 1) |
| ReLU | ReLU |
| Conv2 (256 chan., 4×4 kernels, stride 1, pad 1) | Conv2Transp(256, 3x3 kernels, stride 1, padding 1, output_padding 0) |
| ReLU | ReLU |
| Conv2 (256 chan., 4×4 kernels, stride 2, pad 1) | Conv2Transp(128, 3x3 kernels, stride 2, padding 1, output_padding 1) |
| ReLU | ReLU |
| Linear(512) (both $\mu$ and $\sigma$) | Conv2Transp(128, 3x3 kernels, stride 1, padding 1, output_padding 0) |
| ReLU (both $\mu$ and $\sigma$) | ReLU |
| Linear(K + L) (both $\mu$ and $\sigma$) | Conv2Transp(3, 3x3 kernels, stride 2, padding 1, output_padding 1) |
| - | Sigmoid |

Table 4: The architecture of both the encoder and decoder that make up the CVAE used in the experiment using the CIFAR-10 dataset.

## A.2: Dataset statistics

The number of train samples and test samples for each dataset used in this study is visible in table 5.

| | MNIST | MNIST 3 8 | Fashion MNIST 0,3,4 | CIFAR 10 |
|---|---|---|---|---|
| Number of samples train set | 60,000 | 11,982 | 18,000 | 50,000 |
| Number of samples test set | 10,000 | 1,984 | 3,000 | 10,000 |

Table 5: The architecture of both the encoder and decoder that make up the CVAE used in the MNIST and Fashion MNIST experiments.

## Appendix B: Additional results

### B.1: MNIST dataset with digits 3 and 8

The accuracy of the CNN after training was 99,8% on the testset. This suggests that the model performs close to flawlessly, making it a good model to apply the framework to. After training the CVAE, the sampling process and the varying of these samples resulted in figure 8.

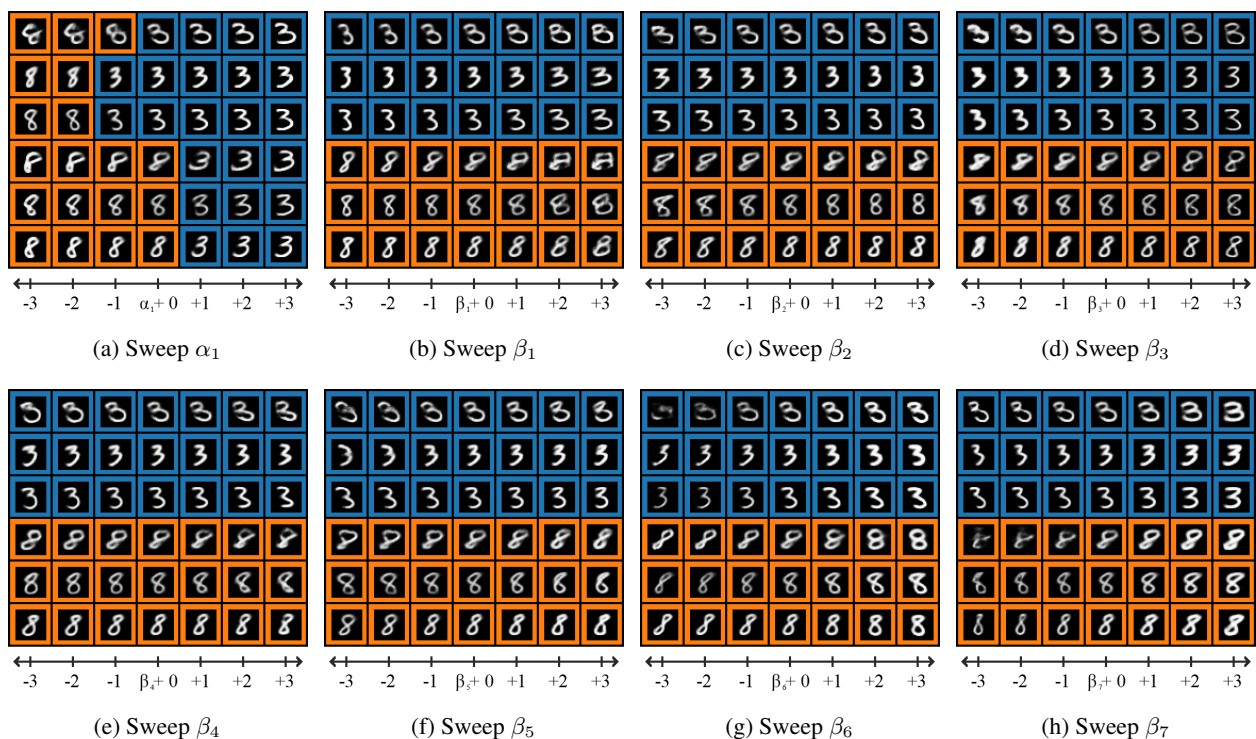

Figure 8: The full results of the attempt to reproduce figure 1. In this figure, the sweeping process of $\beta_4, \beta_5, \beta_6$ and $\beta_7$ and the consequent changes in the generation and classification process are also shown.

### B.2: Complete MNIST dataset

After the training process of the CNN model, the model yielded an accuracy of 99,1% on the testset. As the dataset is more complex compared to the previous experiment, this decrease in performance is expected. Applying the framework around the classifier resulted in figure 9.

### B.3: Fashion MNIST with tops, dresses and coats

The complete figure of the results by the original authors on the Fashion MNIST experiment is shown in figure 10. The complete recreation of our results are displayed in figure 11. These complete versions also include $\beta_3$ and $\beta_4$.

### B.4: Poor model analysis

The classifier that was designed to perform poorly had an accuracy of 52,8% on the testset. This makes the classifier ideal to test the capabilities of the framework on. The extended version of figure 6 is presented in figure 12.

### B.5: CIFAR-10 results

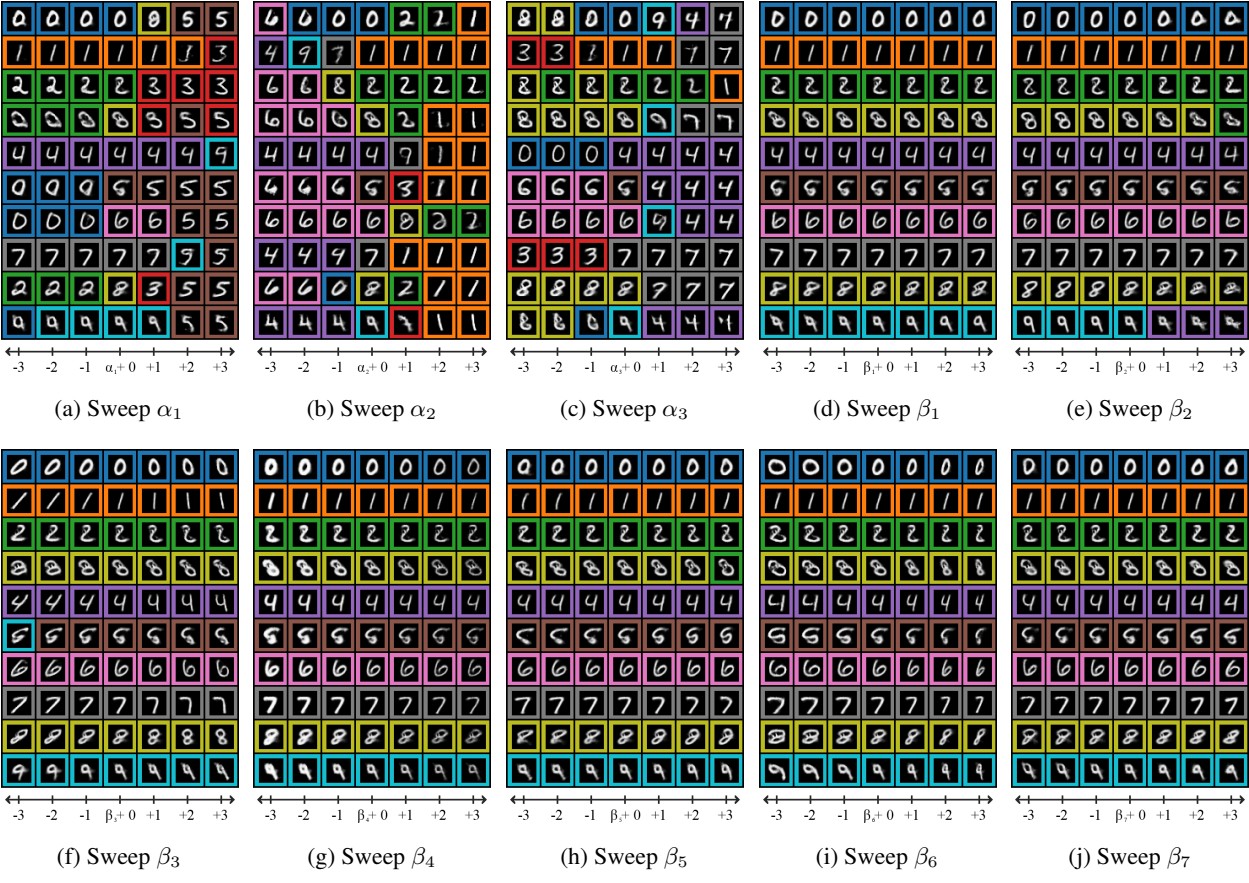

Figure 9: The full results of changing a specific latent factor while keeping the others fixed, on the complete MNIST dataset. In this figure $\alpha_1, \alpha_3, \beta_2, \beta_3, \beta_4$ and $\beta_5$ are also shown.

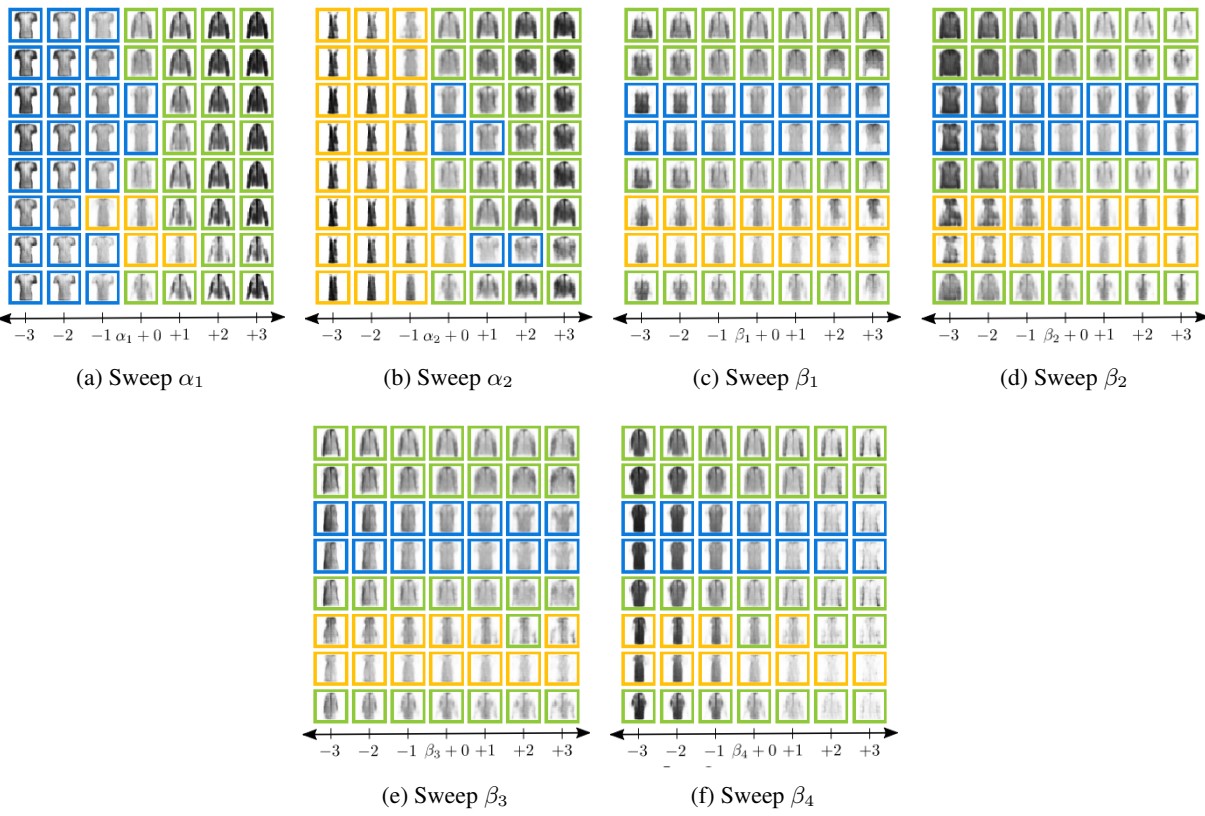

(a) Sweep $\alpha_1$      (b) Sweep $\alpha_2$      (c) Sweep $\beta_1$      (d) Sweep $\beta_2$

(e) Sweep $\beta_3$      (f) Sweep $\beta_4$

Figure 10: The complete version of figure 17 from the original paper.

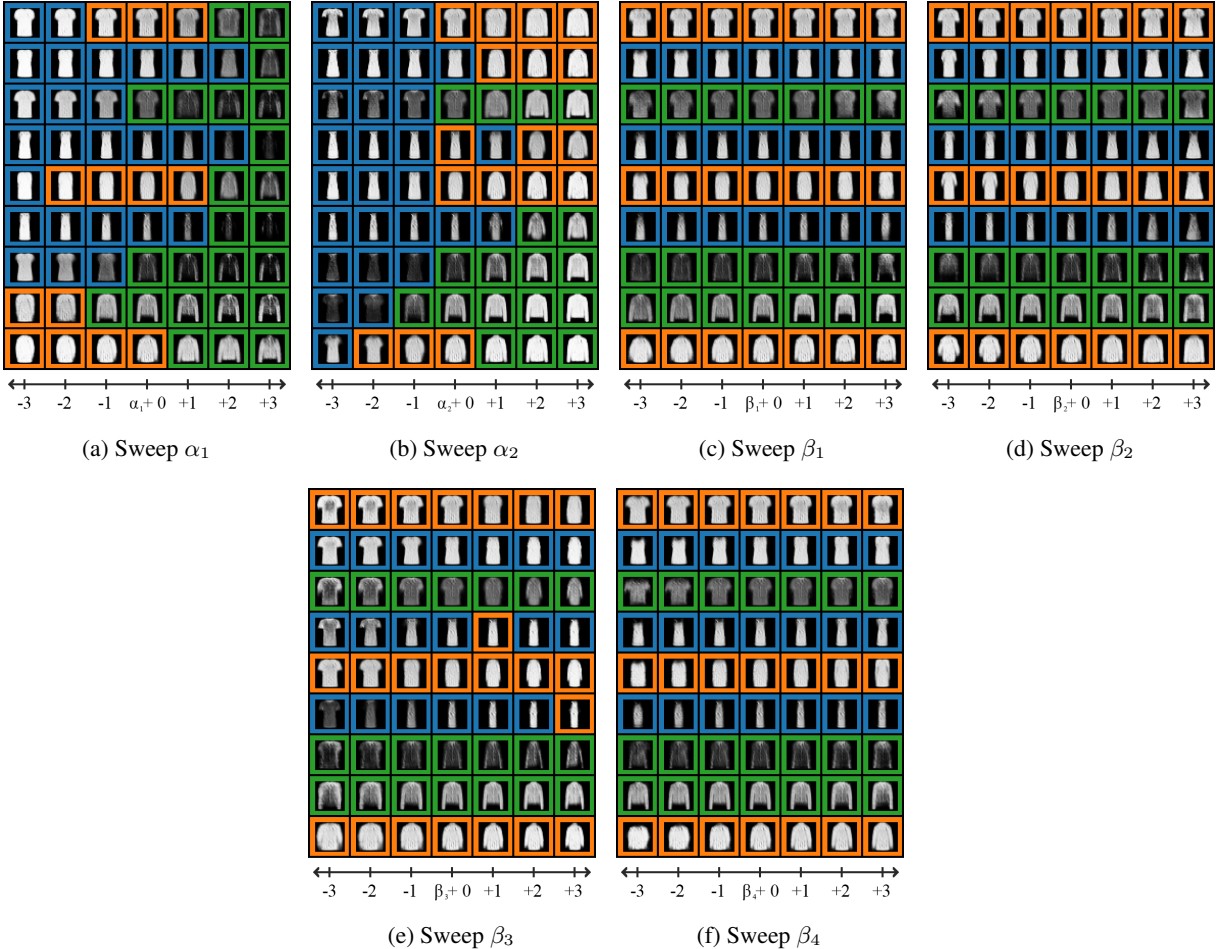

(a) Sweep $\alpha_1$    (b) Sweep $\alpha_2$    (c) Sweep $\beta_1$    (d) Sweep $\beta_2$

(e) Sweep $\beta_3$    (f) Sweep $\beta_4$

Figure 11: The complete version of the results after recreating figure 17 from the original paper.

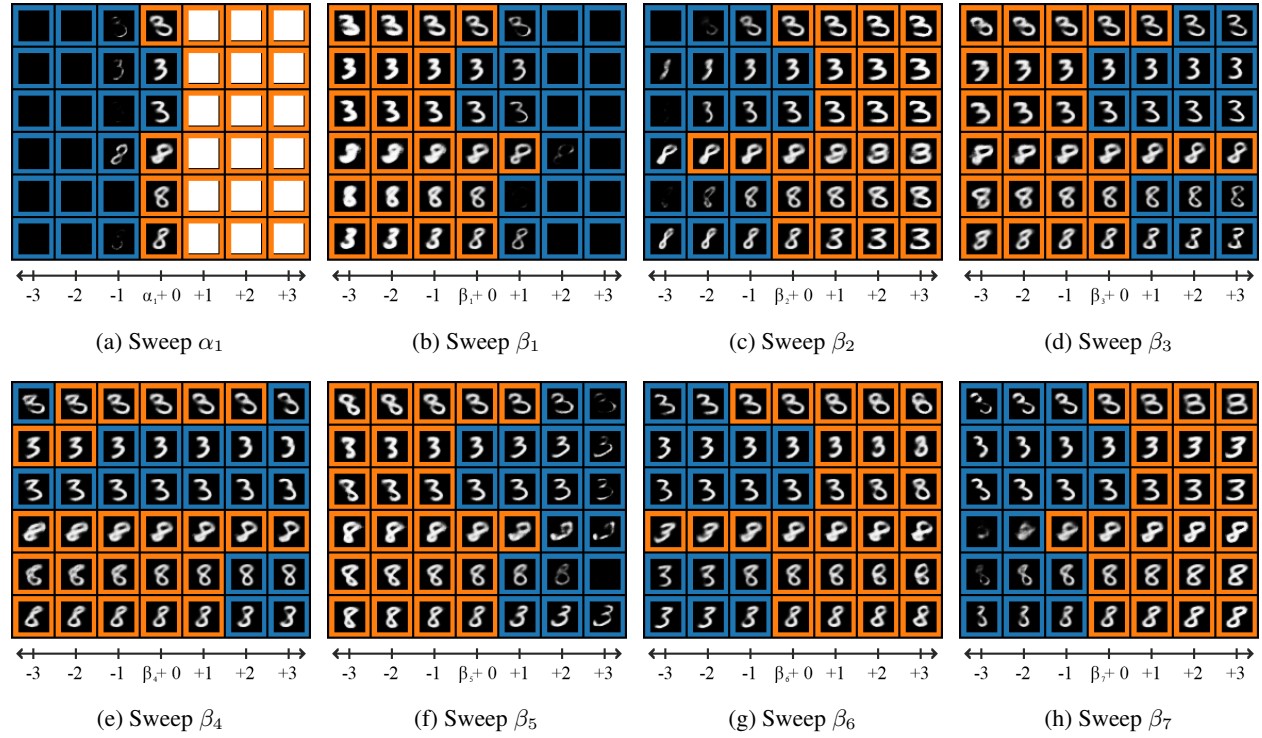

Figure 12: The full figure containing all latent variables, after training with the poorly performing classifier. In this figure, the sweeping process of $\beta_4, \beta_5, \beta_6$ and $\beta_7$ and the consequent changes in the generation and classification process are also shown.

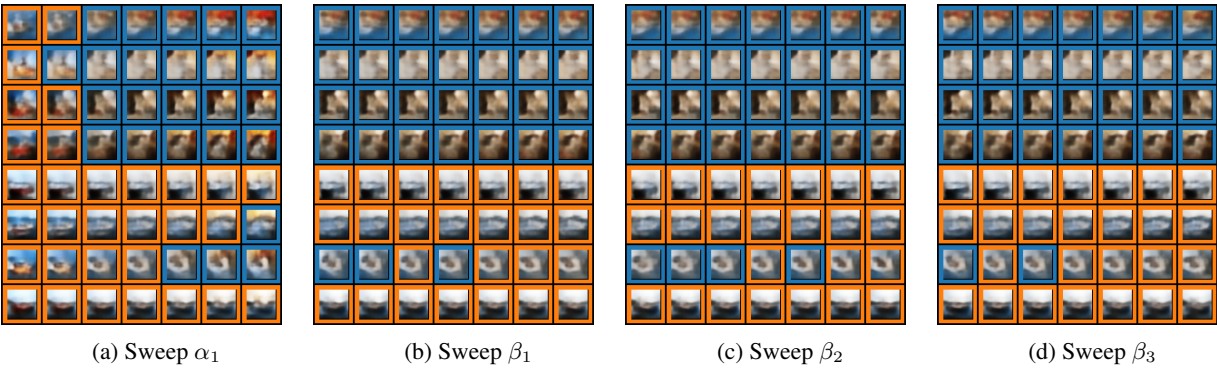

Figure 13: The alpha and some beta values of the model trained on classes "3" and "8" of the CIFAR-10 dataset.

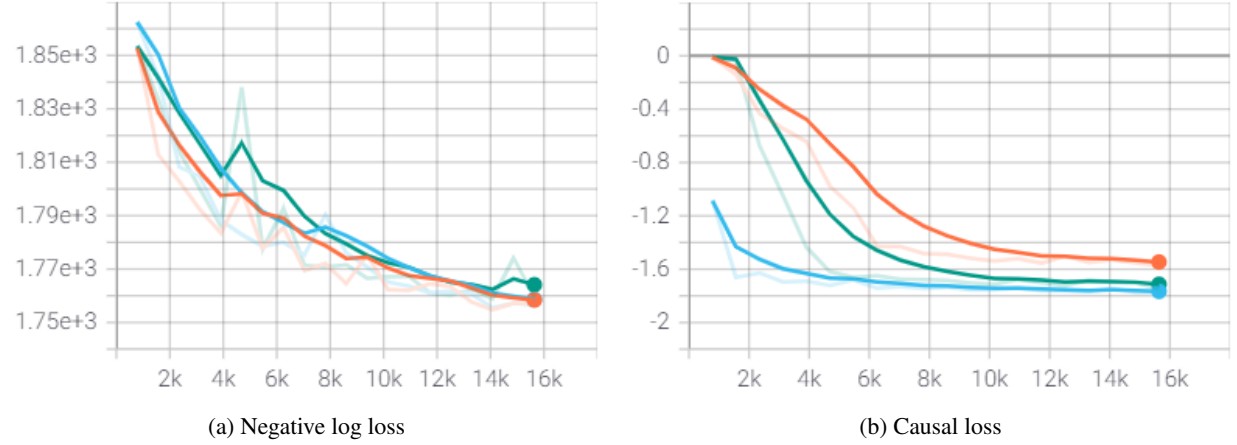

(a) Negative log loss

(b) Causal loss

Figure 14: Validation loss curves for different values of lambda: orange: $\lambda = 0.5$, green: $\lambda = 0.275$, blue: $\lambda = 0.05$

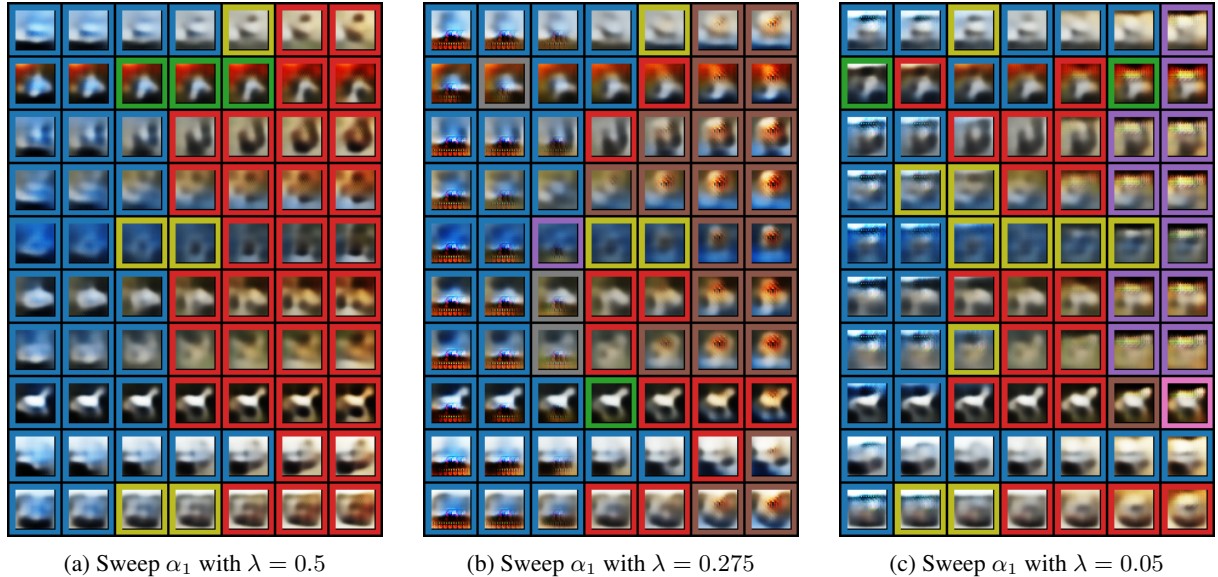

(a) Sweep $\alpha_1$ with $\lambda = 0.5$      (b) Sweep $\alpha_1$ with $\lambda = 0.275$      (c) Sweep $\alpha_1$ with $\lambda = 0.05$

Figure 15: The results of the CIFAR-10 dataset at epoch 5 with different lambdas.

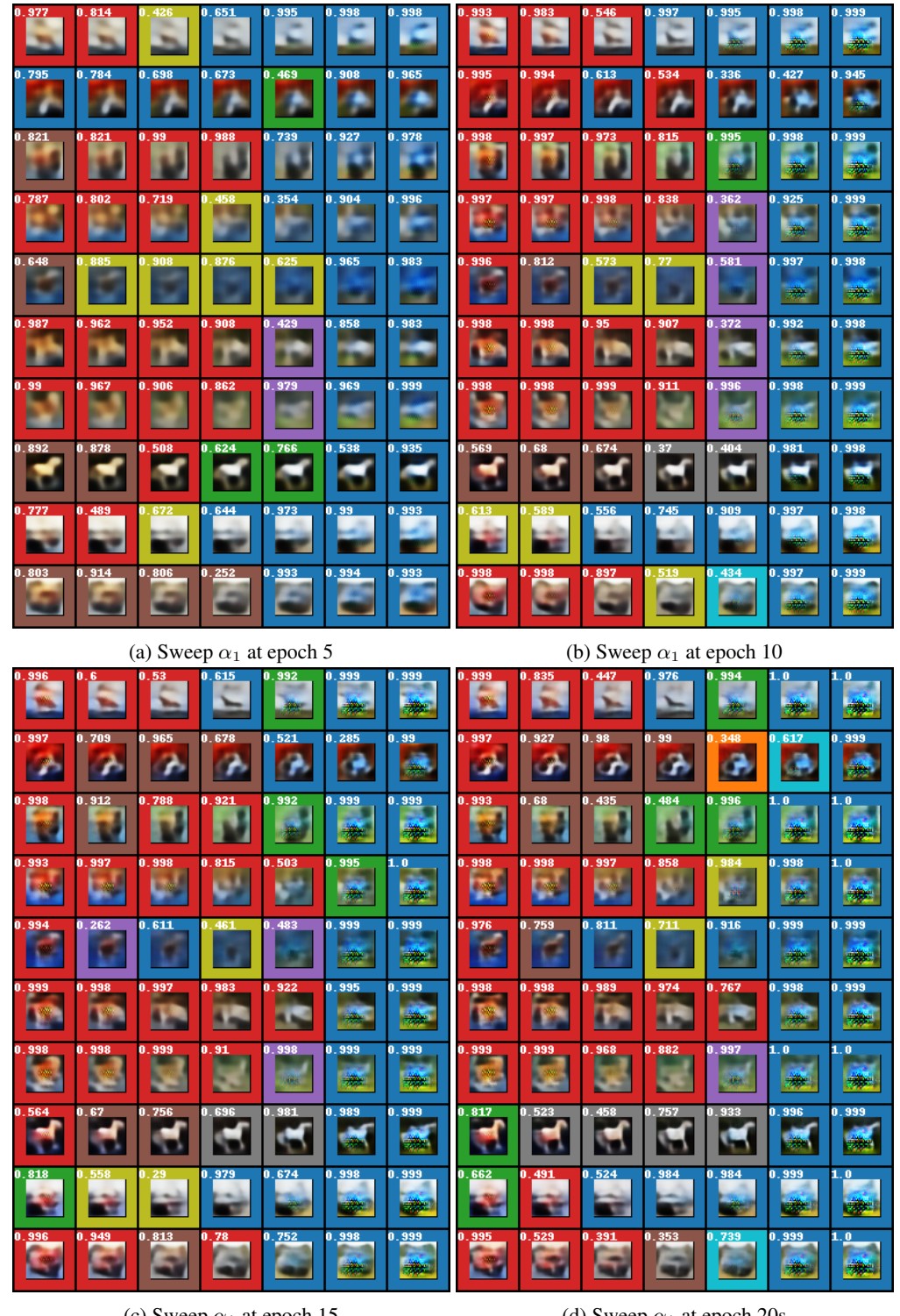

(a) Sweep $\alpha_1$ at epoch 5

(b) Sweep $\alpha_1$ at epoch 10

(c) Sweep $\alpha_1$ at epoch 15

(d) Sweep $\alpha_1$ at epoch 20s

Figure 16: The results of the CIFAR-10 dataset with one $\alpha$-value and $\lambda = 0.5$ with probabilities shown.

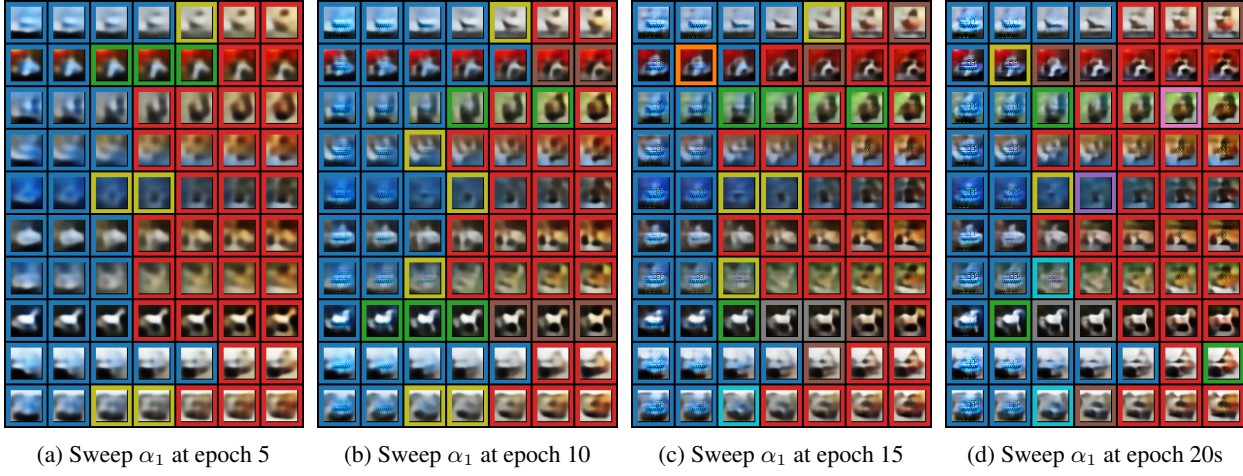

(a) Sweep $\alpha_1$ at epoch 5  (b) Sweep $\alpha_1$ at epoch 10  (c) Sweep $\alpha_1$ at epoch 15  (d) Sweep $\alpha_1$ at epoch 20s

Figure 17: The results of the CIFAR-10 dataset with one $\alpha$-value and $\lambda = 0.5$.

