# OpenReview forum: "Reproducibility study:Generative causal explanations of black-box classifiers"
_ML_Reproducibility_Challenge/2020 — Reject_

### Official Review · AnonReviewer3 · 2021-03-01
**Causal explanations of black-box classifiers  - reproducibility report**

**Rating:** 7
**Confidence:** 3

**Review:**

The authors of this report have tried to reproduce the results of the paper "Generative causal explanations of black-box classifiers" by O’Shaughnessy et.al. They have written up a Reproducibility Summary which has all the required elements. The scope of reproducibility is well-defined, and the report is well-structured and well-written.

The authors have reportedly re-written major portions of the code themselves and have not contacted the original paper's authors. They have shared the link to the new codebase in the report which seems to have sufficient documentation. They have explained the idea in the original paper well in their Introduction. It is impressive that they have not just tried to reproduce the results in the original paper but have also tested the framework on the entire MNIST dataset and the CIFAR-10 dataset, and included a discussion on the results on these datasets. They have even tried to find causal relationships in a low-performance classifier on the MNIST dataset to extend the usefulness of the explanatory framework.
They rightly question the usefulness of the explanations as some of the latent factors cannot be easily interpreted, especially on complex data sets. They also try to explain the scenarios when the framework doesn't work as expected in disentangling the causal and non-causal latent factors.

A few suggestions to improve the report:
1. The authors should include the range of hyperparameters that they have tried before landing on the ones that they have used.
2. The authors should include the training times and the hardware they have used.
3. The authors rightly mention that it was difficult to quantify the results and the judgment of the results was based more on intuition, subject to personal biases and interpretations. Do they have suggestions on what kind of metrics can be employed here? Consulting with the original paper's authors may have led to some good discussions about this aspect and possibly some useful metrics.

In all, this is an impressive reproducibility report. The authors have done due diligence in attempting to reproduce the results and have gone beyond by experimenting on more data sets and models.

**Familiar With The Original Paper:**

I have read the original paper

**Reproducibility Summary:**

Report has summary

---

### Official Review · AnonReviewer1 · 2021-03-05
**Review #1**

**Rating:** 7
**Confidence:** 4

**Review:**

The authors reproduced the results in the paper "Generative causal explanations of black-box classifiers". Though source codes are open-sourced by the original authors, they claimed difficulties of using the original codes and re-implemented most of the codes. Beyond that, they experimented with more datasets and analyzed the experiment results. They also proposed to evaluate the proposed framework to check its ability to examine poor models.

The report is clearly written with detailed analysis. The authors did substantial work to re-implement most of the framework. They discussed the challenges in using the hyper-parameter search methods proposed in the origin paper, i.e. it's too expensive for large models.

Overall, this report clearly described the effort they made to reproduce the results. The ablation study is well motivated with reasonable results.

**Familiar With The Original Paper:**

I have read the original paper

**Reproducibility Summary:**

Report has summary

---

### Decision · Program_Chairs · 2021-03-31

**Decision:**

Reject

**Comment:**

Overall reviews and/or the paper content not good enough for the AC to recommend to the journal.